# Digital Game-Based Heritage Education: Analyzing the Potential of Heritage-Based Video Games

Daniel Camuñas-García [1],*, María Pilar Cáceres-Reche [2], María de la Encarnación Cambil-Hernández [1] and Manuel Enrique Lorenzo-Martín [2]

1 Department of Didactics of Social Sciences, University of Granada, 18011 Granada, Spain; ncambil@ugr.es
2 Department of Didactics and School Organization, University of Granada, 18011 Granada, Spain; caceres@ugr.es (M.P.C.-R.); profesor.manuel.lorenzo@gmail.com (M.E.L.-M.)
* Correspondence: danielcg@ugr.es

**Abstract:** Video games have emerged as a promising tool for enhancing engagement with cultural heritage. However, there is limited knowledge about how existing games can effectively fulfill this role. This study compiled and analyzed 100 video games based on cultural heritage by adapting an existing framework that includes 10 game attributes for a comparative statistical analysis of the games' engagement features. These features include narrative-driven gameplay, information capsules, task-driven learning, ease of play, identity-driven content, open-world exploration, people-centered perspectives, meaningfulness, simulation, and verisimilitude. The analysis revealed that while the majority of games adhered to most of the recommended attributes, features such as task-driven learning, open-world exploration, and simulation were either uncommon or rare. These findings highlight a significant opportunity to develop games that incorporate these underrepresented features, thereby providing more immersive and engaging experiences in heritage education. Furthermore, the study offers a systematic overview of existing games that represent cultural heritage, serving as a valuable resource for developers, designers, and educators in this field.

**Keywords:** video games; gamification; digital game-based learning; cultural heritage; heritage education

## 1. Introduction

Heritage education plays a pivotal role in preserving, transmitting, and celebrating the cultural identities, traditions, and histories of societies across generations. It fosters a sense of belonging, identity, and connection to one's cultural roots, thereby enriching individuals' lives and promoting intercultural understanding and appreciation [1,2]. Furthermore, heritage education acts as a vital tool in safeguarding both tangible and intangible cultural heritage, ensuring the preservation of valuable traditions, practices, and artifacts for future generations [3]. Engaging with cultural heritage allows individuals to deeply appreciate the diversity of human experiences and the interconnectedness of global cultures. This engagement fosters respect, empathy, and tolerance, laying the foundation for the development of a more harmonious and inclusive society [4,5].

To address the growing need and interest in promoting widespread and meaningful engagement with cultural heritage, innovative methods are being adopted. These methods strive to underscore its significance in ways that ignite public interest, understanding, and participation [6]. Among these approaches, digital game-based learning stands out by repurposing games for educational objectives beyond mere entertainment [7]. Several gaming initiatives have successfully garnered attention and facilitated connections between individuals and cultural heritage [8–10]. Furthermore, leading companies in the gaming industry are increasingly looking to weave elements of cultural heritage into their games, indicating a significant shift toward educational integration [11].

Video games are increasingly acknowledged for their significant cultural value, serving as interactive platforms that enable players to explore, engage with, and immerse

themselves in diverse cultural landscapes and narratives [12]. By incorporating historical contexts, artifacts, folklore, and traditions into game narratives, developers provide players with more than mere entertainment; they offer enriching cultural experiences and learning opportunities [13]. This integration of gaming and cultural heritage not only augments the gaming experience, but also plays a pivotal role in preserving and promoting diverse cultural identities and histories worldwide.

However, the majority of heritage-focused games have not undergone systematic analysis to evaluate their potential in promoting engagement with and education about cultural heritage [14]. This oversight has led to a significant gap in our understanding of their effectiveness and overall impact.

Given these considerations, there is a clear need in the literature for an up-to-date, comprehensive, and systematic review of the potential of digital games to promote heritage education. This study aims to bridge this gap by addressing the research question: "Which game design features supporting cultural heritage engagement are present in existing video games, according to the 10 attributes defined in the cultural heritage engagement framework through video games proposed by Camuñas-García et al. [15]"? After conducting a systematic online search, we identified 100 heritage-based computer and mobile games for analysis (a complete list is provided in Appendix A), employing the aforementioned framework for our evaluation.

## 2. Digital Games and Heritage Education

Digital heritage, which involves the creation of digitally constructed landscapes rich in cultural legacy, has emerged as a significant application of video game technology [16,17]. This field has attracted considerable attention for its role in utilizing gaming technology for cultural preservation, especially in archiving 3D models of monuments and historical sites [18]. The cultural sector has made substantial investments in employing video game technology to develop digital heritage tools, and reciprocally, the gaming industry has shown a keen interest in cultural heritage [19]. The process of creating games set in historical periods or at heritage locations requires the collection of extensive data on artistic, architectural, and historical elements, thereby fostering a symbiotic relationship between these two domains [20]. However, a significant critique in this area is the tendency of games to prioritize visual appeal over historical accuracy and authenticity [21,22]. This conflict between authenticity and mass appeal has led experts and heritage professionals to explore alternatives to mainstream games such as educational 'serious games' and mixed-method approaches [23,24].

Commercial games, which primarily aim at entertainment and broad audience appeal, often integrate cultural heritage elements to lend authenticity to the worlds in these games [25]. The cultural sector's growing interest in these games is driven by their significant impact on the cultural economy and tourism [26]. Popular video games that feature cultural heritage sites and artifacts can amplify the public awareness of these locations and items. For instance, the *Assassin's Creed* series, an action-adventure saga set against realistic historical backdrops, explores various places and times with a pronounced emphasis on cultural context. Three of its installments—*Origins*, *Odyssey*, and *Valhalla*, set in ancient Egypt, Greece, and the Viking era, respectively—feature a 'Discovery Tour' mode. This mode enables players to explore the game world at their leisure, diving deep into history and cultural heritage through guided tours and narratives provided by historians and experts [27,28]. Ubisoft has engaged in collaborations with public entities, utilizing its game engine and models for exhibitions such as the one dedicated to Palmyra at the Institut du Monde Arabe in Paris [29]. Monteriggioni (Tuscany, Italy), showcased in the second chapter of *Assassin's Creed*, exemplifies how in-game representation of cultural heritage sites can significantly enhance tourism [30]. Between January and June 2010, the town experienced a 7.24% increase in arrivals and a 16.28% increase in overnight stays compared to the same period in 2009, with the town's Armoury Museum and wall walkways witnessing a 30% rise in admissions in 2010. Surveys conducted by the Monteriggioni municipality

indicated that 11.4% of visitors were influenced by the video game. Inspired by their virtual experiences, many tourists visit real-world locations to relive their in-game adventures or to compare the game's depiction with reality. This inclination, known as game transfer phenomena [31], illustrates how gaming elements can influence real-world perceptions and behaviors, fostering a desire to physically explore these sites [32].

The integration of cultural heritage into mainstream video games significantly contributes to its valorization, thus enhancing the public's recognition of both heritage sites and artifacts [33,34]. This influence is not limited to tangible heritage, but also extends to cultural routes (for instance, as depicted in *Uncharted 2: Among Thieves*) and intangible heritage (for example, *Venba*, a game that provides players with an immersive experience of Tamil culture through cooking).

Aligned with the modern concept of edutainment [35], cultural institutions are incorporating playful elements into their offerings to employ more engaging storytelling and language techniques [36]. Edutainment, which blends education with entertainment to explore the synergy between learning and play, has emerged as a central topic in extensive discussions. Heritage-based video games play a vital role in attracting younger audiences to cultural learning. By immersing players in virtual environments that represent historical and cultural settings, these games act as interactive gateways to the past [37,38]. They not only entertain, but also educate, enabling players to explore and interact with different eras, civilizations, and cultural narratives. This approach makes learning about history and heritage both engaging and relevant, particularly for younger generations accustomed to digital interactivity [39]. Moreover, this medium allows them to develop a deeper understanding and appreciation of the world's diverse cultural heritage, often inspiring further exploration and study beyond the game [40,41].

The growing trend of incorporating video games into exhibitions and museums represents a potentially lasting shift that could significantly shape the development of cultural audiences [42,43]. It is crucial for museums to explore innovative engagement methods while maintaining their cultural and curatorial standards [44]. Video games have emerged as a compelling method to attract and deeply engage younger demographics. Recently, video games and gamification have played a key role in enriching cultural heritage across various regions, often with the support of local authorities [45]. An example of this is the game *Nubla*, developed in collaboration with the Thyssen-Bornemisza National Museum [46]. This game invites players to immerse themselves in modern art through interactive narratives and puzzles inspired by the museum's collection. Beyond attracting new visitors, video games also enhance the overall experience and deepen the understanding of a location's cultural significance and context [47,48].

Serious games, designed to be both engaging and educational, prioritize informative purposes over mere entertainment. They are increasingly utilized across various applications including the showcasing and exploration of cultural heritage [49–51]. These games not only serve as tools for academics and organizations focused on heritage, but also captivate a wider audience. For instance, *Valete Vos Viatores* immerses players in the Roman Empire, educating them on ancient inscription techniques and facilitating interactions with elements of Roman society as a "scriptor titulorum". The development of this game, partially sponsored by the University of Navarra and created in collaboration with research institutions, seeks to cater to both educational objectives and the general public. Despite its focus on heritage and rich dialogs, the game has faced challenges in attracting a broad audience beyond academic circles, highlighting the difficulty in balancing educational content with mass appeal [52].

Furthermore, digital games often encompass intangible heritage. A prominent example is *The Elder Scrolls V: Skyrim*, set in a fantasy world deeply inspired by pre-Christian Scandinavian culture. This game, abundant in references to real-world cultures, depicts a landscape evocative of Scandinavia, albeit within a mythical setting populated by dragons and magic. It serves as a narrative vessel for the heritage of the landscape and ancient traditions, illustrating the vast potential of video games for cultural representation [53].

Another noteworthy example is Barcelona's CaixaForum, a prestigious cultural center renowned for its dedication to integrating contemporary cultural forms with educational initiatives. From 2021 to 2022, CaixaForum Barcelona hosted the 'Homo Ludens' exhibition, which ignited significant discussions on the interplay between video games and contemporary culture. Designed as an educational resource, the exhibition sought to deepen the understanding of various themes within games. Its innovative approach, coupled with the burgeoning interest in video games, was pivotal to the exhibition's success [54]. This favorable outcome encouraged the CaixaForum management team to extend the exhibition to additional locations across the country, recognizing its potential in fostering cultural education and attracting a diverse audience.

This brief overview underscores the dynamic interplay between video games and cultural heritage, showcasing the potential of digital games to preserve and promote diverse cultural heritages. As highlighted in the introduction, the majority of heritage-based games have yet to undergo systematic analysis to ascertain their efficacy in fostering engagement with and education about cultural heritage. Conducting such an analysis could provide invaluable insights for scholars, developers, and educators by enhancing their understanding of how digital games can effectively engage audiences with cultural heritage. It could also help identify the games that exert the most significant impact on heritage education and public engagement initiatives. Subsequently, we will delineate the search and selection process and the criteria employed for the analysis.

## 3. Methods

This study analyzed 100 digital games that are either based on or inspired by the heritage and culture of various countries. These games are accessible online and playable on multiple platforms including computers (Windows and macOS), mobile devices (Android and iOS), and consoles (PlayStation, Xbox, and Nintendo Switch). The methodology employed encompassed two primary steps: (a) search and selection, and (b) content analysis.

### 3.1. Search and Selection

The search was conducted between May and July 2023, with an update in February 2024. It employed a mixed-method search strategy. Initially, a Google search was used to identify well-known heritage-based video games such as the *Assassin's Creed Discovery Tour*. Subsequently, ChatGPT-4 was utilized to generate a list of nonviolent games centered on heritage and culture from various parts of the world. Finally, a manual search was performed across several video game storefronts including Steam, GOG, Itch.io, Google Play Store, App Store, Nintendo eShop, Microsoft Store, and PlayStation Store. These platforms were chosen for their popularity and comprehensive game catalogs, encompassing location-based games, mixed-reality games, and augmented reality (AR) games. The initial Google search uncovered relevant websites such as Games for Change and databases, for example, 50 Games Like, where users can find games similar to those they already enjoy, leading to the discovery of additional titles. In ChatGPT, the query 'make a list of nonviolent video games based on heritage and culture' was entered. This process was repeated until the suggested games began to recur. The search via Google and ChatGPT was concluded upon the realization that no new games were being identified, while on other platforms, all search results were meticulously reviewed.

The initial screening of results for inclusion in the study was conducted by the first researcher, who utilized a combination of methods to evaluate the games. This evaluation involved reviewing official game reviews, watching gameplay videos, and personally playing the games for a duration of up to 30 min. To qualify for consideration in this study, the games needed to meet six specific inclusion criteria:

1.  The game must be classified as a heritage-based video game, drawing on the cultural, artistic, historical, or heritage aspects of a particular region, culture, or era. Its objectives should include the conveyance and preservation of cultural elements through interactive gameplay. This definition excludes simulations that lack game design elements, informational mobile applications, and quiz games such as crosswords, puzzles, anagrams, and memory games.
2.  The game must eschew representations of violence or the inclusion of graphic content that is unsuitable for younger audiences, thereby ensuring it upholds respect for cultural diversity.
3.  The game must be available for download on either a computer or mobile device, without restrictions related to price or platform.
4.  The game must be available in English or must not necessitate language proficiency for gameplay.
5.  The game must have been released within the timeframe from 2013 to 2024.
6.  The game must incorporate themes or cultural elements that span a broad spectrum of cultural heritages including both tangible and intangible aspects. This encompasses folklore, traditional ceremonies, culinary traditions, and personal heritage, depicted through interactions, storytelling, or visual esthetics.

A total of 100 video games satisfied all of the criteria outlined above. Subsequently, we proceeded to analyze them.

### 3.2. Content Analysis

This study analyzed game content by utilizing the 10-attribute cultural heritage engagement framework developed by Camuñas-García et al. [15]. To construct this framework, the authors conducted interviews with 16 scholars and designers involved in the creation of cultural heritage games. Grounded theory was applied to identify pertinent engagement attributes. The 10 identified attributes were subsequently categorized into three dimensions of cultural heritage engagement: cognitive, emotional, and behavioral (see Figure 1). The framework underwent validation through a discussion group with students enrolled in a Heritage Education course, part of the Bachelor's Degree in Primary Education at a teacher training center in Spain. This validation process was further supported by a review of relevant literature from the fields of cultural heritage, education, and game studies.

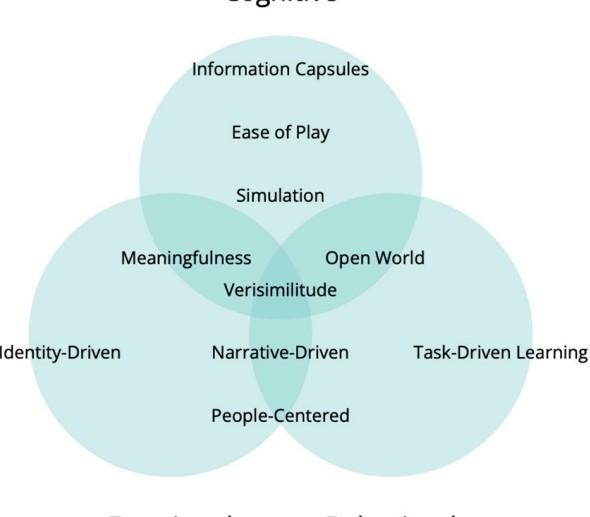

**Figure 1.** Conceptual framework for cultural heritage engagement through video games proposed by Camuñas-García et al. [15].

In May 2023, a comprehensive search was conducted on Scopus to ensure comprehensive coverage of the existing analysis tools. The search string applied was: TITLE-ABS-KEY (game OR gamification) AND (heritage) AND (model OR framework OR theory). This search yielded 685 records; however, none of these records provided detailed methodologies for analyzing cultural heritage engagement through video games. The most relevant finding was a framework designed to assess the level of heritage content in games [55]. Other frameworks, with which we were already familiar [56], primarily focused on the design and evaluation of heritage-based serious games.

In this study, we employed the framework as a checklist. The primary researcher, possessing expertise in both game design and research, coupled with over two decades of regular video game play, assigned a binary value (yes/no) to each of the 10 attributes for every game assessed. Consequently, the analyst's perspective uniquely integrates three distinct interactions with games: as a user, a creator, and a scholar. The descriptions of the 10 attributes are as follows:

1.  Narrative-Driven: Emphasizes a well-structured and coherent narrative as the foundation of the player's experience, crucial for effective information delivery.
2.  Information Capsules: Utilizes discrete content elements including text, images, audio, or interactive components to deliver specific information, thereby enabling learning in manageable increments.
3.  Task-Driven Learning: Engages players in learning through specific in-game tasks and challenges, thereby encouraging the practical application of knowledge and skills.
4.  Ease of Play: Ensures the game is intuitive and user-friendly as well as accessible to players of all skill levels, thereby facilitating a smooth educational experience without undue frustration.
5.  Identity-Driven: Enables players to adopt specific roles or identities within the game, thereby fostering empathy, understanding, and a personal connection to the content.
6.  Open World: Provides a nonlinear environment for gameplay, affording players the autonomy to explore and learn at their own pace, thereby discovering content through various sequences or paths.
7.  People-Centered: Centers the gaming experience around human interactions, whether with game characters or other players, thereby highlighting the significance of relationships and communication in the learning process.
8.  Meaningfulness: Strives to provide experiences that are not only entertaining, but also profound and impactful, potentially altering the player's viewpoint or expanding their comprehension.
9.  Simulation: Employs realistic simulations to mirror real-world situations, phenomena, or systems, thereby granting players the opportunity to learn within a controlled yet realistic setting.
10. Verisimilitude: Achieves a high level of authenticity and coherence within the game world, ensuring that, despite the presence of fictional elements, the environment resonates as real and credible to the player.

To support the validity and reliability of the analysis, the study employed various verification strategies. These strategies included detailing the analyst's background (as previously mentioned), engaging extensively in gameplay to gain a deep familiarity with the games, enriching the playing experience with additional data sources such as reviews and gameplay videos, and consistently revisiting the results throughout the analysis and writing process to ensure a uniform interpretation of the assigned values. Upon finalizing the list of games and assigning values to each attribute for every game, the data were analyzed using SPSS software, version 27.

## 4. Results

The attributes in the content analysis were categorized from 'Very Common' to 'Rare' based on their prevalence. Beginning with 'Very Common' attributes, which a significant majority of games exhibited, 'Ease of Play' was present in 96% of games, and 'Narrative-

Driven' in 90%. 'Identity-Driven' and 'People-Centered' followed, featured in 85% and 89% of games, respectively. These attributes underscore the industry's focus on accessibility, storytelling, and personal engagement, reflecting a commitment to creating more inclusive and accessible experiences for players from diverse backgrounds and with various interests. Moving to the 'Common' attributes, 'Information Capsules' and 'Verisimilitude' were each present in 68% of games, while 'Meaningfulness' appeared in 60%. This suggests a notable yet moderate inclusion of educational content, realistic portrayals, and impactful experiences designed to enrich the players' engagement with cultural heritage. The 'Uncommon' category included 'Task-Driven Learning', present in 25% of the games, suggesting a targeted use of task-based educational methods to foster learning and interaction with cultural content. Finally, 'Open World' and 'Simulation', categorized as 'Rare', were found in only 15% and 21% of games, respectively. This underscores these features as the least adopted in cultural heritage-themed games, likely due to complexities or specific design philosophies favoring narrative and identity elements over open exploration or detailed simulation mechanics. Table 1 and Figure 2 illustrate the percentage of games incorporating each attribute, alongside descriptive statistics such as mean, standard deviation, median, minimum, and maximum values, regarding the number of attributes observed across the games.

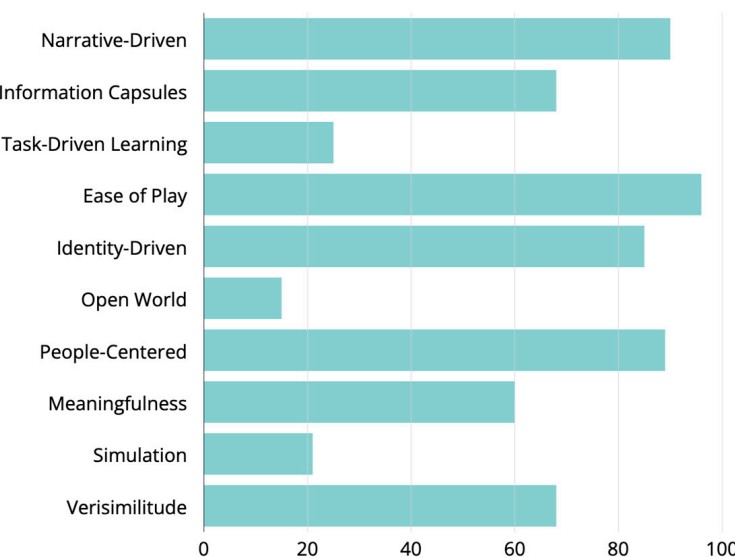

**Figure 2.** Percentage of games by presence of individual engagement attributes.

**Table 1.** Percentage of games by presence of individual engagement attributes.

| Attributes | Overall (*n* = 100) | | | | |
|---|---|---|---|---|---|
| Narrative-Driven | 90 | | | | |
| Information Capsules | 68 | | | | |
| Task-Driven Learning | 25 | | | | |
| Ease of Play | 96 | | | | |
| Identity-Driven | 85 | | | | |
| Open World | 15 | | | | |
| People-Centered | 89 | | | | |
| Meaningfulness | 60 | | | | |
| Simulation | 21 | | | | |
| Verisimilitude | 68 | | | | |
| | **M** | **SD** | **Md** | **Min** | **Max** |
| **Overall (*n* = 100)** | 6.8 | 2.43 | 7 | 2 | 10 |

On average, the analyzed games typically exhibited between six and seven of these attributes, with a relatively moderate variance around the mean (SD = 2.43). A median of seven attributes per game further suggests that at least half of the games incorporated a majority of these features.

Likewise, the chi-square test of independence was applied to assess the interrelationships among the attributes. The *p*-values derived from these tests are presented in a correlation matrix (see Figure 3), where values less than the conventional alpha level of 0.05 are indicative of statistically significant associations. It is important to note that Yates' correction was employed for 2 × 2 contingency tables to adjust for continuity, ensuring a more conservative significance test in the presence of small sample sizes. The color gradient from green to blue in the matrix indicates the range of significance, with green representing nonsignificance ($p \geq 0.05$) and blue denoting significance ($p < 0.05$). The diagonal of the matrix is marked with an 'X', signifying the redundancy of testing a variable against itself.

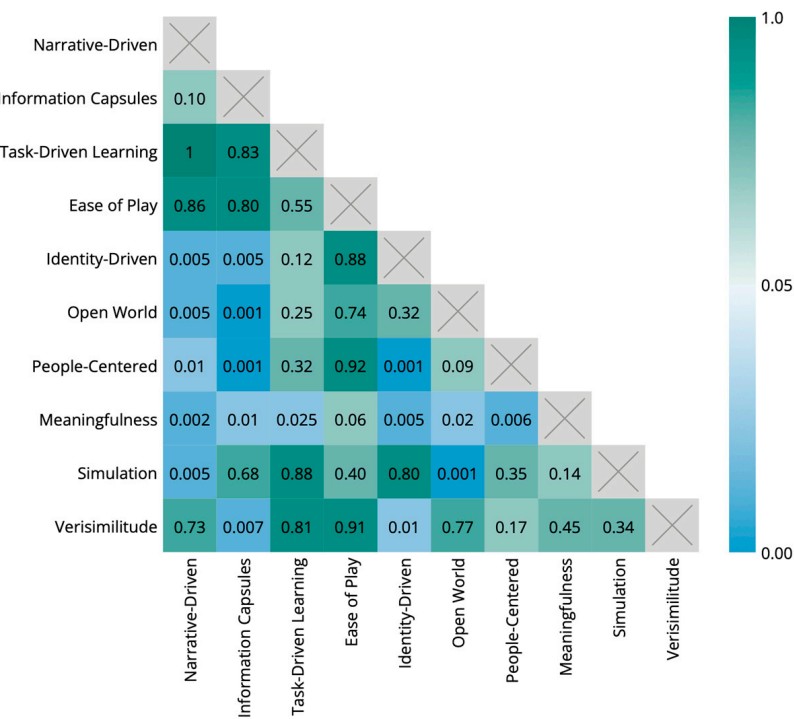

**Figure 3.** Correlation matrix of *p*-values from the chi-square test of independence for heritage engagement attributes.

A significant correlation was observed between 'Information Capsules' and both the 'Open World' and 'People-Centered' attributes. This relationship suggests that the manner in which information is presented ('Information Capsules') is intricately linked with the player's autonomy to explore ('Open World') and the prominence of interactions with nonplayer characters (NPCs) ('People-Centered'). Furthermore, a significant correlation was found between the 'Identity-Driven' and 'People-Centered' attributes, implying that games emphasizing elements of personal identity often incorporate a people-centered perspective. This synergy may serve as a fundamental principle in the development of these types of games, ensuring that they effectively communicate cultural stories while also resonating with players on a personal level, reflecting their own identities and experiences. Finally, another significant correlation was observed between the 'Open World' and 'Simulation' attributes, pointing to a substantial interplay between the freedom of exploration in an open world and the realistic representation of environments in simulations within cultural heritage games. This suggests that games offering both extensive exploration and simulations that closely mirror real-world scenarios can provide a more immersive and engaging experience, particularly in terms of historical accuracy.

Based on the framework proposed by Camuñas-García et al. [15], we identified four significant attributes affecting the cognitive dimension, five impacting the emotional dimension, and three influencing the behavioral dimension (see Table 2).

**Table 2.** Very common and common attributes by cultural heritage engagement dimension.

| Attributes | Cognitive | Emotional | Behavioral |
|---|---|---|---|
| Narrative-Driven | | X | X |
| Information Capsules | X | | |
| Ease of Play | X | | |
| Identity-Driven | | X | |
| People-Centered | | X | X |
| Meaningfulness | X | X | |
| Verisimilitude | X | X | X |

## 5. Discussion

Our study, analyzing the engagement potential of video games with respect to cultural heritage issues, resides at the intersection of heritage education and game-based learning research. It focuses on game artifacts, their messages, and their impacts on players. The existing literature has primarily concentrated on evaluating serious games and simulations [57–59] as well as the authenticity of representations within games [60,61]. Our research contributes new insights by compiling and analyzing the most comprehensive collection of games in this field to date. Whereas previous studies have categorized existing serious games and outlined their affordances using theoretical frameworks, our approach employs a content analysis framework that is topic-specific and empirically developed. Our findings reveal that while most video games display a majority of the engagement attributes, some attributes are uncommon or rare. In the following sections, we explore how each attribute is represented within our corpus of games and its relationship to the related literature.

### 5.1. Findings on Heritage Engagement Attributes

#### 5.1.1. Very Common Attributes

The 'Ease of Play' attribute was highly prevalent, observed in 96% of the analyzed games. Although related to accessibility, it is crucial to distinguish between the two concepts. 'Ease of Play' refers to the intuitiveness and straightforwardness of a game's interface and controls, whereas accessibility focuses on making games playable and enjoyable for individuals with a broad spectrum of disabilities including those affecting vision, hearing, mobility, or cognition [62,63]. In most games we analyzed, 'Ease of Play' extended beyond the initial learning curve to encompass the entire user experience, ensuring that players of all skill levels could achieve success and enjoyment without excessive frustration. This aspect included the implementation of carefully designed difficulty settings, the presentation of fair challenges, and the clear outlining of goals and objectives [64]. For example, the game *Jotun* features dynamic difficulty adjustment, a feature that automatically modifies the game's challenge level according to the player's skill, ensuring a consistently engaging experience for both novices and experienced players alike.

'Narrative-Driven' gameplay was evident in the majority of the analyzed games, accounting for 90%. This emphasis on narrative manifested in various forms. Many games, such as *Valiant Hearts: The Great War* and *Detective Di: The Silk Rose Murders*, draw from deeply rooted historical narratives, traditions, and the real-world histories of the cultures they depict. Others like *Röki*, *A Highland Song*, and *Raji: An Ancient Epic* delve into the folklore and mythology of various cultures, bringing to life the myths, legends, and folktales passed down through generations. Furthermore, titles such as *Assemble With Care* and *Hindsight* explore personal and family histories, immersing players in stories deeply connected to the characters' heritage. Games like *Pentiment* and *What Remains of Edith Finch* employ nonlinear storytelling techniques, allowing the players' choices and actions

to significantly influence the narrative outcome. These narratives do more than engage; they play a pivotal role in constructing knowledge and motivating learners, highlighting the profound impact of storytelling on engagement and learning [65,66].

Games featuring a 'People-Centered' design, which constituted 89% of the analyzed titles, often incorporate mechanisms to enhance collaboration, dialog, and relationship development between players and nonplayer characters (NPCs). Titles such as *Alba: A Wildlife Adventure* and *Signs of the Sojourner* feature systems that enable players to make meaningful choices. These choices impact not only the game world and its inhabitants, but also foster a sense of agency and empathy among players [67]. By integrating mechanics that require communication, negotiation, and occasionally compromise, these games effectively mirror real-world social interactions [68]. This approach enriches the gaming experience by infusing it with the authenticity and depth of interpersonal connections, thus underscoring the profound impact of 'People-Centered' design in creating engaging and meaningful gaming experiences.

'Identity-Driven' design was a significant characteristic observed in 85% of the analyzed games. This design approach encompasses character customization, narrative choices reflecting the player's personal values or decisions, and game mechanics that adapt in response to the player's identity-related choices [69,70]. In these games, characters are portrayed not merely as protagonists, but also as friends, family members, or anthropomorphized animals, engaging in human-like interactions, as seen in *Night in the Woods*. Adopting this approach enhances the learning experience by fostering a deeper connection between the player and the explored cultural narratives. 'Identity-Driven' elements create a more immersive and personalized gaming experience, allowing players to see reflections of their own values and decisions within the game world, thus enhancing engagement and deepening narrative involvement [71].

### 5.1.2. Common Attributes

In 68% of the analyzed games, 'Information Capsules' served a pivotal role, functioning as either narrative or gameplay elements. These capsules deliver succinct lore, instructions, or items to players, significantly enhancing their understanding and immersion in the game world. Manifesting in various forms—interactive objects, text boxes, audio logs, or in-game characters—these capsules provide vital information or backstory, enriching the player's experience. In culturally focused games, 'Information Capsules' assume an even more crucial role, offering historical context, cultural details, and educational content that deepens the narrative's richness [72]. A prime example is *Never Alone (Kisima Ingitchuna)*, which uses information capsules to effectively convey the stories, beliefs, and traditions of the Iñupiat people, demonstrating how these elements bridge the gap between entertainment and education.

A total of 68% of the analyzed games were characterized by their high degree of believability, encapsulated in the attribute of 'Verisimilitude.' This term describes how closely games mirror reality or create experiences that feel authentically realistic. For instance, *Season: A Letter to the Future* exemplifies this attribute with its meticulous attention to environmental details, character expressions, and sound design, crafting an immersive, lifelike experience. Similarly, the *Assassin's Creed Discovery Tour* mode presents a carefully constructed world where NPC behaviors, wildlife, and dynamic weather patterns all contribute to the sense of realism [73]. This commitment to realism and verisimilitude not only deepens immersion and player engagement, but also facilitates the integration of real-world knowledge into the gaming environment, making the experience both more enriching and educational [74–76].

'Meaningfulness', identified in 60% of the analyzed games, is intimately connected to the games' ability to foster relationships with family and friends, enhance knowledge and skills, and make positive contributions to society, thereby fostering a vision of a better world [77]. These games prompt players to reflect on the impact of history on contemporary and future societal norms and values, cultivating a sense of responsibility and promoting

global citizenship. A notable example is *Before I Forget*, where players navigate the protagonist's journey through fragmented memories to unveil their life story. This exploration enriches the gaming experience with deeper layers of meaning and emotion, prompting reflection on personal life experiences and relationships, thus enhancing understanding of the human condition and emphasizing the significance of cherishing memories and connections [78].

### 5.1.3. Uncommon Attributes

A significant majority (75%) of the analyzed games prioritized story elements, exploration, and player-driven choices, indicating a strong preference for narrative depth and interactive storytelling. In contrast, the remaining 25% were classified as 'Task-Driven', emphasizing gameplay mechanics such as puzzle-solving, resource management, and simulation-based activities. These games engage players in problem-solving, strategic planning, and the management of in-game resources or environments to facilitate progression [79,80]. *Heaven's Vault* exemplifies the balance between these two types of gameplay. Players take on the role of archeologist Aliya Elasra, exploring ancient ruins within the enigmatic Nebula. The game's procedurally generated sites provide unique adventures in each playthrough, while deciphering an ancient language uncovers the history of lost civilizations. Through engaging in puzzle-solving, translating inscriptions, and making decisions that influence the narrative, players are immersed deeply in both the storyline and the tasks. *Heaven's Vault* illustrates that the key to achieving a balanced game design lies in seamlessly integrating narrative depth with task-driven gameplay, thus enriching the overall player experience [81].

### 5.1.4. Rare Attributes

Although only 21% of video games were classified within the 'Simulation' genre, their significant contribution to experiential learning cannot be overstated. In these games, players do more than merely engage with simulations; they interact with sophisticated models that replicate real-world processes such as economic systems, ecological environments, or historical events [82]. These systems within the games react to player decisions, thereby deepening engagement and vividly demonstrating how varied choices lead to diverse outcomes. This is exemplified by *The Oregon Trail*, which challenges players to manage resources and make strategic decisions reflective of a historical journey, thus mirroring the real-world impacts of such choices. By navigating complex systems and reflecting on the consequences of their actions, players develop critical thinking skills. This engagement not only enhances their understanding of the modeled systems, but also enriches their learning experience by demonstrating the real-world applicability of their decisions, thereby underscoring the profound educational value of simulation games.

'Open World' games, constituting 15% of the analyzed titles including examples such as Call of the Sea and RiME, provide players with expansive, nonlinear environments for exploration and interaction. This genre affords players a profound sense of freedom and agency, enabling them to forge their own paths and shape their experiences within the game world [83,84]. The open-world format is particularly conducive to exploration and discovery, essential elements of engaging with cultural heritage. In these games, players are not simply passive recipients of cultural information; rather, they actively seek it, assembling historical and cultural narratives through the environment, artifacts, and interactions with nonplayer characters (NPCs). This active engagement not only fosters a deeper and more personal connection with the depicted cultural heritage, but also enhances the educational value of the gaming experience by allowing players to immerse themselves in and learn from the cultures and histories intricately woven into the game world.

*5.2. Implications and Further Research*

This article contributes to the multidisciplinary effort of engaging individuals with cultural heritage, specifically through research exploring the potential role of video games in promoting cultural heritage engagement. By addressing the guiding question of this study, we aim to provide a deeper understanding of the current landscape of games in relation to cultural heritage, identifying both prevalent engagement features and existing gaps. For professionals in heritage education and communication, this study offers a detailed analysis of digital games for potential application. For instance, they might select games that offer comprehensive engagement or those that emphasize specific attributes. For game developers, we highlight underutilized engagement attributes such as task-driven learning, open-world exploration, and simulation that offer opportunities for innovation. Additionally, our review of how various features are incorporated into existing games serves as a contemporary reference for the design of new cultural heritage games.

Our findings underscore the necessity for a broader approach in subsequent research that involves incorporating virtual reality (VR) games, examining games beyond the scope of the scientific literature, and undertaking empirical studies to assess the games analyzed within the context of heritage education practices. We advocate for future investigations to evaluate the impacts of the examples discussed in this article and to explore the potential relationship between engagement attributes and empirical outcomes. By scrutinizing uncommon and rare engagement attributes, particularly those that are challenging to implement, future studies can more effectively determine the merit of a concerted effort to utilize these attributes more extensively. Furthermore, a separate analysis of representations of tangible versus intangible heritage could unveil potential variations in engagement with these two categories.

*5.3. Limitations*

While this list strives to be as comprehensive as possible, it may not encompass the full spectrum of cultural heritage themes present within the video game industry. Games containing violent content were intentionally excluded to prioritize educational and positive cultural engagement. However, this exclusion should not be interpreted as a disregard for their cultural value or significance. Indeed, certain games that incorporate violence such as *Red Dead Redemption 2* provide deep cultural narratives and representations, significantly contributing to cultural heritage engagement. Moreover, the emphasis on games released in the past 11 years was a deliberate decision to capture the latest trends and advancements in the video game industry, along with the contemporary portrayal of cultural heritage within these titles. This methodological approach enabled the study to capitalize on the significant technological and narrative developments of the recent decade.

Other limitations pertain to the methodologies employed. Firstly, the content analysis was conducted by a single researcher, who brings the perspective of both an experienced player and a designer. As outlined in the Methods (Section 3), multiple strategies were utilized to ensure the integrity and quality of the analysis. Secondly, while this study explored the potential of games for engagement, it did not examine the actual real-world impacts of these games. The potential for engagement identified may not necessarily translate into tangible outcomes, influenced by a myriad of factors beyond the games' design, a consideration that falls outside the purview of this paper.

## 6. Conclusions

This study analyzed 100 cultural heritage games accessible on both mobile and computer devices. An analysis of cultural heritage engagement revealed that most games adhered to the majority of expert recommendations. However, attributes such as task-driven learning, open-world exploration, and simulations were uncommon or rare. Game developers might consider focusing on these underrepresented engagement attributes to empirically investigate their effects on player engagement. The study also suggests that while cultural heritage games engage players with heritage through various gameplay

experiences, the wide variety of games in the sample indicates that future research should broaden its scope. This could uncover previously overlooked opportunities for engagement and education through digital game-based cultural heritage.

**Author Contributions:** Conceptualization, D.C.-G. and M.d.l.E.C.-H.; Methodology, D.C.-G.; Validation, D.C.-G., M.P.C.-R. and M.d.l.E.C.-H.; Formal Analysis, D.C.-G.; Investigation, D.C.-G., M.P.C.-R. and M.d.l.E.C.-H.; Resources, D.C.-G. and M.E.L.-M.; Data Curation, D.C.-G.; Writing—Original Draft Preparation, D.C.-G., M.P.C.-R., M.d.l.E.C.-H. and M.E.L.-M.; Writing—Review and Editing, D.C.-G., M.P.C.-R., M.d.l.E.C.-H. and M.E.L.-M.; Visualization, D.C.-G.; Supervision, M.P.C.-R. and M.d.l.E.C.-H.; Funding Acquisition, D.C.-G. and M.P.C.-R. All authors have read and agreed to the published version of the manuscript.

**Funding:** This research is part of the thesis project by D.C.-G, entitled "Video Games as an Educational Tool for Teaching Cultural Heritage", supported by a predoctoral fellowship from the Spanish Ministry of Universities (FPU20/00281). Additionally, this research contributes to the project "Influence of Video Game Addiction on the Adolescent Population of Andalusia in Educational and Family Contexts (PRY127/22)", funded by the Andalusian Studies Centre Foundation (CENTRA).

**Institutional Review Board Statement:** Not applicable.

**Informed Consent Statement:** Not applicable.

**Data Availability Statement:** Data are contained within the article.

**Conflicts of Interest:** The authors declare no conflicts of interest.

### Appendix A

**Table A1.** List of the 100 video games based on cultural heritage analyzed along with their attributes.

| Games | Released | Developer | (1) * | (2) | (3) | (4) | (5) | (6) | (7) | (8) | (9) | (10) |
|---|---|---|---|---|---|---|---|---|---|---|---|---|
| 80 Days | 2014 | Inkle | Yes | Yes | No | Yes | Yes | No | Yes | Yes | No | No |
| A Highland Song | 2023 | Inkle | Yes | Yes | No | Yes | Yes | No | Yes | Yes | No | Yes |
| Alba: A Wildlife Adventure | 2020 | Ustwo Games | Yes | Yes | Yes | Yes | Yes | Yes | Yes | Yes | Yes | Yes |
| A Life in Music | 2019 | TuoMuseo | Yes | No | No | Yes | Yes | No | Yes | No | No | Yes |
| Ambition: A Minuet in Power | 2021 | Joy Manufacturing Co. | Yes | Yes | No | Yes | Yes | No | Yes | Yes | No | No |
| Animalario | 2018 | EducaThyssen | No | Yes | No | Yes | Yes | No | Yes | No | No | Yes |
| A Perfect Day | 2019 | Perfect Day Studio | Yes | Yes | Yes | Yes | Yes | No | Yes | Yes | No | Yes |
| A Place for the Unwilling | 2019 | AlPixel Games | Yes | Yes | No | Yes | Yes | No | Yes | Yes | No | No |
| A Space for the Unbound | 2023 | Mojiken Studio | Yes | Yes | Yes | Yes | Yes | No | Yes | Yes | No | Yes |
| Assassin's Creed Discovery Tour Bundle | 2021 | Ubisoft | No | Yes | No | Yes | No | Yes | Yes | No | Yes | Yes |
| Assemble With Care | 2019 | Ustwo Games | Yes | Yes | No | Yes | Yes | No | Yes | Yes | Yes | Yes |
| Before I Forget | 2020 | 3-Fold Games | Yes | Yes | No | Yes | Yes | No | Yes | Yes | No | Yes |
| Before Your Eyes | 2020 | GoodbyeWorld Games | Yes | Yes | No | Yes | Yes | No | Yes | Yes | No | No |
| Behind the Frame: The Finest Scenery | 2021 | Silver Lining Studio | Yes | Yes | No | Yes | Yes | No | Yes | Yes | No | Yes |
| Beyond Our Lives | 2018 | TuoMuseo | Yes | Yes | No | Yes | Yes | No | Yes | No | No | Yes |
| Broken Sword 5: The Serpent's Curse | 2013 | Revolution Software | Yes | Yes | No | Yes | Yes | No | Yes | No | No | Yes |
| Burly Men at Sea | 2016 | Brain&Brain | Yes | No | No | Yes | Yes | No | Yes | Yes | No | No |
| Call of the Sea | 2020 | Out of the Blue Games | Yes | Yes | No | Yes | Yes | Yes | Yes | No | Yes | Yes |
| Carto | 2020 | Sunhead Games | Yes | Yes | No | Yes | Yes | Yes | Yes | No | Yes | Yes |
| Chants of Sennaar | 2023 | Rundisc | Yes | No | No | Yes | Yes | No | Yes | Yes | Yes | No |

**Table A1.** *Cont.*

| Games | Released | Developer | (1) * | (2) | (3) | (4) | (5) | (6) | (7) | (8) | (9) | (10) |
|---|---|---|---|---|---|---|---|---|---|---|---|---|
| Chicory: A Colorful Tale | 2021 | Wishes Ultd. | Yes | Yes | Yes | Yes | Yes | Yes | Yes | Yes | No | No |
| Crowns and Pawns: Kingdom of Deceit | 2022 | Tag of Joy | Yes | Yes | No | Yes | Yes | No | Yes | Yes | No | Yes |
| Detective Di: The Silk Rose Murders | 2019 | Nupixo Games | Yes | Yes | No | Yes | Yes | No | Yes | Yes | No | Yes |
| Dordogne | 2023 | Un Je Ne Sais Quoi | Yes | Yes | Yes | Yes | Yes | No | Yes | Yes | No | Yes |
| Embracelet | 2020 | Machineboy | Yes | Yes | Yes | Yes | Yes | Yes | Yes | No | No | Yes |
| Essays on Empathy | 2021 | Deconstructeam | Yes | Yes | No | Yes | Yes | No | Yes | Yes | No | Yes |
| Father and Son | 2017 | TuoMuseo | Yes | Yes | No | Yes | Yes | No | Yes | No | No | Yes |
| Father and Son 2 | 2022 | TuoMuseo | Yes | Yes | No | Yes | Yes | No | Yes | No | No | Yes |
| Gerda: A Flame in Winter | 2022 | PortaPlay | Yes | Yes | Yes | Yes | Yes | No | Yes | Yes | No | Yes |
| Goetia | 2016 | Sushee | Yes | No | No | No | No | No | No | No | No | Yes |
| Goetia 2 | 2022 | Moeity | Yes | No | No | Yes | No | No | No | No | No | No |
| Heaven's Vault | 2019 | Inkle | Yes | Yes | Yes | Yes | Yes | Yes | Yes | Yes | Yes | Yes |
| Herald: An Interactive Period Drama | 2017 | Wispfire | Yes | Yes | No | Yes | Yes | No | Yes | Yes | No | Yes |
| Hindsight | 2022 | Team Hindsight | Yes | Yes | No | Yes | Yes | No | Yes | Yes | No | Yes |
| Hourglass | 2021 | Cyberwave | No | No | No | Yes | No | Yes | No | No | No | No |
| How We Know We're Alive | 2023 | Motvind Studios | Yes | Yes | No | Yes | Yes | No | Yes | Yes | No | Yes |
| I Am Dead | 2020 | Hollow Ponds | Yes | No | No | Yes | Yes | No | Yes | Yes | No | No |
| Inua: A Story in Ice and Time | 2022 | The Pixel Hunt | Yes | Yes | No | Yes | Yes | No | Yes | Yes | No | Yes |
| Jotun | 2015 | Thunder Lotus Games | Yes | No | No | Yes | No | No | No | Yes | No | No |
| League of Enthusiastic Losers | 2021 | Yookond. | Yes | Yes | No | Yes | Yes | No | Yes | Yes | No | Yes |
| Lost Words: Beyond the Page | 2020 | Sketchbook Games | Yes | No | Yes | Yes | Yes | No | Yes | Yes | No | No |
| Luna: The Shadow Dust | 2020 | Lantern Studio | Yes | Yes | Yes | Yes | Yes | No | Yes | Yes | No | No |
| Lushfoil Photography Sim | 2024 | Matt Newell | No | No | No | Yes | Yes | Yes | No | Yes | Yes | Yes |
| Milkmaid of the Milky Way | 2017 | Machineboy | Yes | Yes | No | Yes | Yes | No | Yes | Yes | No | Yes |
| Mr Tic Toc & the Endless City | 2018 | Un Je Ne Sais Quoi | Yes | No | No | Yes | Yes | No | Yes | No | No | No |
| Mulaka | 2018 | Lienzo | Yes | Yes | Yes | Yes | Yes | No | Yes | Yes | No | No |
| Mythwrecked: Ambrosia Island | 2024 | Polygon Treehouse | Yes | Yes | Yes | Yes | Yes | No | Yes | Yes | No | No |
| Never Alone (Kisima Ingitchuna) | 2014 | E-Line Media | Yes | Yes | No | Yes | Yes | No | Yes | Yes | No | Yes |
| Night in the Woods | 2017 | Infinite Fall | Yes | Yes | No | Yes | Yes | No | Yes | Yes | No | Yes |
| Nubla | 2015 | EducaThyssen | Yes | Yes | No | Yes | Yes | No | Yes | No | No | No |
| Nubla 2 | 2019 | EducaThyssen | Yes | Yes | No | Yes | Yes | No | Yes | No | No | No |
| Occupy White Walls | 2018 | Kultura Ex Machina | No | No | No | Yes | No | Yes | Yes | No | Yes | No |
| Old Man's Journey | 2017 | Broken Rules | Yes | Yes | No | Yes | Yes | No | Yes | Yes | No | Yes |
| Once Upon a Sea | 2020 | Blimey | Yes | No | No | Yes | No | No | No | No | No | Yes |
| Open Roads | 2024 | Open Roads Team | Yes | Yes | No | Yes | Yes | No | Yes | Yes | No | Yes |
| Over the Alps | 2019 | Stave Studios | Yes | Yes | Yes | Yes | Yes | No | Yes | Yes | No | Yes |
| Past for Future | 2018 | TuoMuseo | Yes | Yes | No | Yes | Yes | No | Yes | No | No | Yes |

**Table A1.** *Cont.*

| Games | Released | Developer | (1) * | (2) | (3) | (4) | (5) | (6) | (7) | (8) | (9) | (10) |
|---|---|---|---|---|---|---|---|---|---|---|---|---|
| Pentiment | 2022 | Obsidian | Yes | Yes | No | Yes | Yes | No | Yes | Yes | No | Yes |
| Plasticity | 2019 | Plasticity Games | Yes | No | No | Yes | Yes | No | Yes | Yes | Yes | Yes |
| Raji: An Ancient Epic | 2020 | Nodding Heads Games | Yes | No | No | Yes | Yes | No | Yes | No | No | No |
| Reliefs: The Time of the Lemures | 2023 | Calepin Studio | Yes | No | No | Yes | No | Yes | No | No | Yes | Yes |
| RiME | 2017 | Tequila Works | Yes | No | No | Yes | Yes | Yes | No | No | Yes | No |
| Röki | 2020 | Polygon Treehouse | Yes | Yes | Yes | Yes | Yes | No | Yes | Yes | No | No |
| Sadhana: The Way Back | 2020 | La Générale de Production | Yes | No | No | Yes | No | No | Yes | Yes | No | No |
| Season: A Letter to the Future | 2023 | Scavengers Studio | Yes | Yes | No | Yes | Yes | Yes | Yes | Yes | Yes | Yes |
| Signs of the Sojourner | 2020 | Echodog Games | Yes | Yes | No | Yes | Yes | No | Yes | Yes | Yes | No |
| Snufkin: Melody of Moominvalley | 2024 | Hyper Games | Yes | Yes | Yes | Yes | Yes | Yes | Yes | Yes | No | No |
| South of the Circle | 2020 | State of Play | Yes | Yes | No | Yes | Yes | No | Yes | Yes | No | Yes |
| Syberia 3 | 2017 | Koalabs | Yes | Yes | No | No | Yes | No | Yes | No | Yes | Yes |
| Syberia: The World Before | 2022 | Koalabs | Yes | Yes | No | Yes | Yes | No | Yes | Yes | Yes | Yes |
| Tengami | 2014 | Nyamyam | Yes | No | Yes | Yes | Yes | No | Yes | Yes | No | No |
| The Book of Distance | 2020 | National Film Board of Canada | Yes | No | No | Yes | Yes | No | Yes | Yes | No | Yes |
| The Dead Sea Scrolls Adventure | 2022 | Cave3 Studios | Yes | Yes | No | Yes | Yes | No | Yes | Yes | No | Yes |
| The Dedal Games | 2021 | EducaThyssen | Yes | No | No | Yes | No | No | Yes | No | No | No |
| The Excavation of Hob's Barrow | 2022 | Cloak and Dagger Games | Yes | Yes | Yes | Yes | Yes | No | Yes | Yes | No | Yes |
| The Forever Labyrinth | 2024 | Inkle | Yes | Yes | No | Yes | No | No | Yes | No | No | No |
| The House of Da Vinci | 2017 | Blue Brain Games | No | No | No | Yes | No | No | No | No | Yes | No |
| The Master's Pupil | 2023 | Pat Naoum | Yes | No | No | Yes | Yes | No | Yes | Yes | No | No |
| The Oregon Trail | 2021 | Gameloft | No | Yes | Yes | No | Yes | No | Yes | No | Yes | Yes |
| The Plague Doctor of Wippra | 2022 | Electrocosmos | Yes | Yes | Yes | Yes | Yes | No | Yes | Yes | No | Yes |
| The Procession to Calvary | 2020 | Joe Richardson | Yes | No | No | Yes | No | No | Yes | No | No | No |
| The Rewinder | 2021 | Misty Mountain Studio | Yes | Yes | No | Yes | Yes | No | Yes | Yes | No | No |
| The Stillness of the Wind | 2019 | Memory of God | No | No | No | Yes | Yes | No | Yes | Yes | No | Yes |
| This is a True Story | 2022 | Frosty Pop | Yes | Yes | No | Yes | Yes | No | Yes | Yes | No | Yes |
| TOEM | 2021 | Something We Made | Yes | Yes | Yes | Yes | Yes | No | Yes | Yes | Yes | No |
| Torn Away | 2022 | Perelesoq | Yes | No | Yes | Yes | Yes | No | Yes | Yes | No | Yes |
| Treasures of the Aegean | 2021 | Undercoders | Yes | No | No | Yes | Yes | No | Yes | No | No | No |
| Tukoni | 2020 | Oksana Bula | Yes | Yes | No | Yes | Yes | No | Yes | Yes | No | No |
| Type:Rider | 2013 | Cosmografik | No | Yes | No | Yes | No | No | No | No | No | No |
| Underground Blossom | 2023 | Rusty Lake | Yes | No | No | No | Yes | No | Yes | No | No | No |
| Valete Vos Viatores | 2022 | Trahelium | No | No | Yes | Yes | Yes | Yes | Yes | No | Yes | Yes |
| Valiant Hearts: Coming Home | 2023 | Ubisoft | Yes | Yes | Yes | Yes | Yes | No | Yes | Yes | No | Yes |
| Valiant Hearts: The Great War | 2014 | Ubisoft | Yes | Yes | Yes | Yes | Yes | No | Yes | Yes | No | Yes |
| Vandals | 2018 | Cosmografik | Yes | No | No | Yes | No | No | No | Yes | No | No |

**Table A1.** *Cont.*

| Games | Released | Developer | (1) * | (2) | (3) | (4) | (5) | (6) | (7) | (8) | (9) | (10) |
|---|---|---|---|---|---|---|---|---|---|---|---|---|
| Venba | 2023 | Visai Games | Yes | Yes | No | Yes | Yes | No | Yes | Yes | Yes | Yes |
| Voodoo Detective | 2022 | Short Sleeve Studio | Yes | Yes | Yes | Yes | Yes | No | Yes | Yes | No | Yes |
| What Remains of Edith Finch | 2017 | Giant Sparrow | Yes | No | No | Yes | Yes | No | Yes | Yes | No | Yes |
| When Rivers Were Trails | 2019 | Elizabeth LaPensée | Yes | Yes | No | Yes | Yes | No | Yes | Yes | No | Yes |
| Where the Water Tastes Like Wine | 2018 | Dim Bulb Games | Yes | No | No | Yes | Yes | No | Yes | Yes | No | No |
| Wide Ocean Big Jacket | 2020 | Turnfollow | Yes | Yes | No | Yes | Yes | No | Yes | Yes | No | No |

* (1) Narrative-Driven, (2) Information Capsules, (3) Task-Driven Learning, (4) Ease of Play, (5) Identity-Driven, (6) Open World, (7) People-Centered, (8) Meaningfulness, (9) Simulation, and (10) Verisimilitude.

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
