# Peer review of "Digital Game-Based Heritage Education: Analyzing the Potential of Heritage-Based Video Games"

_education, doi:10.3390/educsci14040396_

Round 1

Reviewer 1 Report

Comments and Suggestions for Authors

The research presented in this manuscript offers a commendable foray into the burgeoning intersection of digital gaming and cultural heritage. The topic's pertinence in a rapidly digitalising world cannot be overstated; hence, the significance of this study lies in its timely exploration of how digital games can be both a medium and a repository for cultural narratives.

The strength of this paper lies in its innovative approach to analysing the content of digital games through a cultural heritage lens. The meticulous construction of a 10-attribute framework for this analysis is laudable, as it sets a tangible parameter within which games are evaluated. The methodological design, grounded in grounded theory and validated through scholarly engagement, stands as a testament to the robust academic foundations upon which this research is built.

However, it is imperative to address a rather significant oversight pertaining to the statistical analysis employed. The use of Pearson's correlation coefficient in the manuscript is misplaced, as it is traditionally reserved for measuring linear relationships between two continuous variables. The binary nature of the data (yes/no) in this context calls for a different approach. The chi-square test of independence or Fisher's exact test, which are better suited to categorical data, would yield more statistically sound insights. This misapplication suggests a misunderstanding of applied statistics that must be corrected to uphold the integrity of the research.

Furthermore, it is advisable to re-evaluate the data using an appropriate measure of association for binary data, ensuring that the analysis is reflective of the categorical nature of the attributes assessed. A revised correlation matrix should employ the lower triangle exclusively to present the results, thereby eliminating redundancy and enhancing clarity. This streamlined presentation would not only be statistically proper but also more accessible to readers, avoiding unnecessary repetition and fostering an efficient conveyance of information.

The endeavour of the authors is both timely and needed. The insights drawn from their research have the potential to significantly contribute to our understanding of cultural heritage within digital platforms. Addressing the noted statistical and linguistic errors will undoubtedly enhance the calibre and clarity of this work, positioning it as a seminal piece within the intersection of gaming and cultural education. In conclusion, with these amendments, the paper promises to be an influential addition to the corpus of literature in this innovative field.

Comments on the Quality of English Language

The document exhibits areas in need of linguistic enhancement. For the most part, the language utilised is of a high standard, yet there are instances where grammatical inconsistencies and deviations from academic conventions are present. These lapses, though not pervasive, do require attention to ensure the manuscript's language meets the exemplary standards expected in scholarly communications. Engaging the services of an experienced academic editor or utilising advanced linguistic software, could rectify these imperfections.

In particular, the areas most affected by these linguistic inconsistencies are often those where complex statistical information is conveyed. This is a crucial juncture where precision in language underpins the clarity of the presented data. Such sections would benefit greatly from a thorough review to ensure the language is as precise and clear as the data it aims to describe.

Author Response

Thank you very much for your comments and suggestions for improvement. They have greatly helped to enhance the quality of the manuscript. Below are the changes made:

1. The grammar and style of the English used throughout the manuscript have been reviewed to improve its clarity and comprehension. The review was carried out by an external native English speaker. Given the number of changes in the text, these have not been highlighted in yellow.

2. The Pearson correlation coefficient has been removed, and the chi-square test of independence with Yates' correction has been used instead. This has been highlighted in yellow in the Results section.

3. The correlation matrix has been corrected with the suggested changes. The new data have been introduced, and only the lower triangle has been used to eliminate redundancies and improve clarity. This has been highlighted in yellow in the Results section.

Once again, thank you very much for your comments and your good collaboration.

Best regards.

Reviewer 2 Report

Comments and Suggestions for Authors
  1. The literature review provides a comprehensive overview of the importance of heritage education and the role of digital games in promoting cultural heritage. It discusses various aspects, including the integration of cultural heritage in popular video games, the concept of serious games, and examples of successful initiatives in museums and exhibitions. The introduction sets up the context well, and subsequent sections delve into the topic with appropriate depth.

  2.  
  3. Inconsistent capitalization of "video games" (sometimes capitalized, sometimes not). These inconsistencies should be addressed for clarity and consistency.

The presentation of results is coherent and well-organized. The text progresses logically from summarizing the prevalence of attributes to analyzing correlations between attributes. This logical flow ensures that readers can follow the analysis without confusion. Furthermore, the use of tables, figures, and a correlation matrix enhances the organization and clarity of the results, making it easier for readers to interpret the data.

The arguments presented in the discussion are compelling, supported by empirical evidence from the content analysis of games. The analysis of each attribute's prevalence and its relationship with related literature adds depth to the discussion, demonstrating the study's theoretical grounding and analytical rigor. Moreover, the discussion goes beyond mere description, offering insights into the educational and experiential implications of different design features. This adds value to the discussion by highlighting the practical significance of the findings for heritage education, game development, and future research directions.

Conclusions are wrapped up into the Discussion. Consider a separate section.

Author Response

Thank you very much for your comments and suggestions for improvement. They have greatly helped to enhance the quality of the manuscript. Below are the changes made:

1. The grammar and style of the English used throughout the manuscript have been reviewed to improve its clarity and comprehension. The review was carried out by an external native English speaker. Given the number of changes in the text, these have not been highlighted in yellow.

2. A Conclusions section has been added to close the manuscript with the most important findings. This has been highlighted in yellow.

Once again, thank you very much for your comments and your good collaboration.

Best regards.

Round 2

Reviewer 1 Report

Comments and Suggestions for Authors

I have reviewed the suggested changes in the document, and I can confirm that they have been duly incorporated. I am satisfied with its current state and believe it meets the necessary standards for submission.

Comments on the Quality of English Language

I am grateful for the thorough English language review you have conducted, which has greatly improved the manuscript's clarity and readability. I acknowledge that while the current draft is considerably enhanced, there remains scope for minor stylistic refinements.